# Does Childhood Maltreatment Predict Moral Disgust? The Underlying Mediating Mechanisms

**DOI:** 10.3390/ijerph191610411

**Published:** 2022-08-21

**Authors:** Qingji Zhang, Yue Zhou, Ziyuan Chen, Yanhui Xiang

**Affiliations:** 1School of Marxism, Dongguan University of Technology, Dongguan 523808, China; 2Department of Psychology, Hunan Normal University, Changsha 410081, China; 3Cognition and Human Behavior Key Laboratory of Hunan Province, Hunan Normal University, Changsha 410081, China; 4Institute of Developmental Psychology, Beijing Normal University, Beijing 100091, China

**Keywords:** childhood maltreatment, moral disgust, emotional intelligence, empathy, structural equation model

## Abstract

Although moral disgust is one of the most important moral emotions, there is limited evidence about the antecedents of it in China. This paper aimed to discuss the linkage between childhood maltreatment and moral disgust, and investigated the specific mechanism between these two variables from the perspective of emotional development and moral development, respectively, based on the Tripartite Model. By combining random sampling and cluster sampling, this study recruited 968 participants from college. Then, childhood maltreatment, moral disgust, emotional intelligence, and empathy were measured separately by using the Childhood Trauma Questionnaire (CTQ), Moral Disgust Scale (MD), Wong Law Emotional Intelligence Scale (WLEIS), and Interpersonal Reactivity Index–C (IRI). Additionally, the results of the mediation model analysis show that childhood maltreatment is negatively predictable of moral disgust. In addition, the mechanism by which childhood maltreatment influences moral disgust could be explained by the effect of emotional intelligence on empathy. To sum up, this study explored and explained the specific mechanism between childhood maltreatment and moral disgust, replenishing previous achievements and providing support for the design of intervention on moral disgust by improving emotional intelligence and empathy.

## 1. Introduction

Disgust is a kind of basic emotion, motivating people to retreat from threats. It originally emerged to help people avoid physical contaminants [1]. Then, with the development of human civilization, this emotion has been applied to social and moral domains, thus giving rise to moral disgust [2]. Moral disgust refers to the experience of disgust when exposed to moral transgressors or offenders, which helps to protect the spiritual self [3,4]. Experiencing this kind of emotion not only motivates people to avoid moral violators, but also urges themselves to observe the societal norm [5,6]. Accordingly, moral disgust is crucial to people’s social adjustment, as well as social order maintenance. Given this, it is worthwhile to explore the antecedents of moral disgust, while relative research is limited.

There may exist many potential factors affecting moral disgust, among which the home environment children experience when they are young may be the root cause. Additionally, childhood maltreatment refers to a seriously disordered parent–child relationship accompanied with an unstable family environment [7], which has been proved by a great amount of literature to be the antecedent of many negative results, such as aggression [8,9], borderline personality disorder [10] and addiction [11]. Importantly, previous studies have found that exposure to maltreatment impairs individuals’ moral sensitivity [12] and result in more immoral behaviours [13]. Considering the harmful effects of child abuse on moral grounds, childhood maltreatment may also play a negative role in one’s development of moral disgust. Thus, we propose **H1**: childhood maltreatment negatively predicts moral disgust. The childhood maltreatment–moral disgust link, however, is not enough to provide support for the design of intervention on moral disgust, as this abuse experience has become an immutable fact. In order to find an effective solution to mitigate the negative effect of childhood maltreatment on moral disgust, the current study needs to further explore the influencing mechanism between them, which still remains unclear. Therefore, the main aim of the present study is to investigate the underlying mediating roles between childhood maltreatment and moral disgust.

Moral disgust, as a moral emotion, is necessarily affected by moral and emotional development. Specifically, emotional development enables individuals to respond to disgusting stimulus with an appropriate emotional reaction, and the development of morality reflects the individual’s possession of moral concepts and moral attitudes [14]. Additionally, the development of morality and emotion depends on a healthy family environment [15,16]. In addition, empirical evidence has supported that childhood maltreatment is detrimental to moral formation and emotional regulation [17,18,19]. Therefore, we suspect that childhood maltreatment may hamper the formation of moral disgust by hindering moral and emotional development. Overall, this study probed into the specific mechanism between childhood maltreatment and moral disgust from the perspective of emotional and moral development based on the tripartite model, so as to enrich relevant theoretical research about moral disgust and provide suggestions to intervene in the development of moral disgust.

From the perspective of emotional development, emotion is not only the source of behaviour dynamics, but it also plays an important role in individual socialization. Some studies have shown that people with high emotional intelligence are more prone to have a keen sense while maintaining a stable standard of value. That characteristic contributes to individuals making better judgments in moral situations and is the basis of moral disgust [20,21]. Emotional intelligence is identified as the ability to operate emotions, therefore, high emotional intelligence groups are possibly more predisposed to processing and regulating emotions well [22,23,24]. Therefore, we hypothesized that the mechanism by which childhood maltreatment influenced moral disgust could be explained by emotional intelligence. As implied by the tripartite model of the impact of the family, the family environment in which the child lives plays an important role in the development of the child’s emotional competence [16]. On the one hand, researchers have proved that people with an abused history are more likely to have difficulties in regulating and processing emotions [25,26], suggesting that childhood maltreatment could affect the function of emotional intelligence. Additionally, relevant research has also provided support that childhood maltreatment could directly leave negative impacts on emotional intelligence [27]. Since childhood is the key period of emotional development, and the growth environment filled with uncontrollability and pain brought by childhood maltreatment may hinder children’s normal development of emotion functioning, thus affecting emotional intelligence [26]. On the other hand, researchers have proved that emotional intelligence has an effect on moral reasoning, which is the process that people make decisions according to their internal standards and values. It shows that emotional intelligence can affect a person’s moral concepts [22], and then may impact their emotional experience on moral events, ultimately affecting moral disgust. Combining the above theoretical and empirical evidence, the adverse family context in childhood leads to low emotional intelligence and consequently low disgust for immorality. Therefore, we proposed **H2**: childhood maltreatment negatively predicts moral disgust by affecting emotional intelligence.

From moral development’s perspective, many psychologists believe that empathy is crucial to the formation of morality [20,28]. Empathy is a kind of affective response according to the perception of other’s emotions, facilitating people to experience what others feel. As a result, people with high empathy could feel the same pain when others suffer, thus having a better comprehension of moral principle and the internalization of moral standards, which are also the foundation of moral disgust [29,30]. In addition, a number of studies have demonstrated that empathy is closely correlated to prosocial behaviour and other moral behaviours [31]. As empathy could be considered as an individual’s affective responsiveness, it also fits the tripartite model’s view of childhood maltreatment affecting the development of emotional competence. Thus, empathy may also explain the link between childhood maltreatment and moral disgust, and some indirect evidence helps support our hypothesis. First, previous literature has shown that childhood maltreatment impedes the development of empathy [10]. This may be accounted for by the negative cognition patterns and attribution style that individuals with an abused history tend to use, which have a negative effect on the ability of reading others’ emotions [32]. As a result, abused children may find it hard to understand the internal emotion behind actions and events, eventually resulting in lower empathy. Secondly, empathy could not only affect moral cognition, but also stimulate a strong emotional motivation to correct immoral behaviours [28,33], which makes people more sensitive to immoral situations, thus arousing stronger moral disgust. Therefore, we proposed **H3**: childhood maltreatment influences moral disgust by affecting empathy.

In addition, emotional intelligence and empathy are two closely related variables, and previous studies have provided supports that they are positively correlated [34,35]. Moreover, research has demonstrated that emotional intelligence can effectively promote empathy, which may be because high emotional intelligence groups can better perceive, comprehend and operate emotions, and then they can better comprehend their own and others’ emotions and the consequences of their behaviours, thus improving their empathy [36]. Therefore, we believe that childhood maltreatment may hamper emotional intelligence, and then, the lower emotional intelligence impedes the foundation of empathy, ultimately contributing to lower moral disgust. Thus, we proposed **H4**: childhood maltreatment negatively predicts moral disgust by affecting empathy through emotional intelligence.

In conclusion, this study aimed to examine the possible association between childhood maltreatment and moral disgust, and investigated the mediating roles of emotional intelligence and empathy, respectively, from the perspective of emotional and moral development based on the tripartite model. The following hypotheses are proposed in this study: (1) childhood maltreatment negatively predicts moral disgust; (2) childhood maltreatment predicts moral disgust negatively through emotional intelligence; (3) childhood maltreatment predicts moral disgust negatively through empathy; and (4) childhood maltreatment influences empathy through emotional intelligence, and then predicts moral disgust.

## 2. Methods

### 2.1. Participants

The sample data in this study were from an ongoing project named “Early Adverse Environment Influences Cognitive Affective Mechanism”. Some data have been used in previous studies [9,27]. Participants were recruited from four universities in mainland China in the method of random sampling and cluster sampling. After excluding data from participants who did not complete the questionnaire or completed it in a mischievous way, data from a total of 968 participants were included in the final analysis.The youngest participant was 17 years old, and the oldest participant was 26 years old. The mean age was 19.07 years old (SD = 1.55), and 64% (n = 619) were women. Participants are invited to complete the questionnaire required for the study, which takes approximately 40 min. Following review of an informed consent form, participants completed the questionnaires. All participants were compensated financially after finishing all the questionnaires. The current study was approved by the Academic Committee of authors’ department.

### 2.2. Measures

#### 2.2.1. Childhood Trauma Questionnaire (CTQ)

The Childhood Trauma Questionnaire was originally developed by Bernstein [37]. In this study, we adopted the Chinese revised version [38], which has 23 items, including emotional abuse, emotional neglect, physical abuse and physical neglect. The items were rated on a 5-point Likert-type scale of 1 (never) to 5 (frequently) that measured the extent to which individuals have been abused when they were children. Sample items included “people in my family said hurtful or insulting things to me.” Additionally, empirical studies conducted in China have supported its reliability and validity [9]. In present study, the internal consistency of the scale was acceptable: 0.65, 0.74, 0.78, 0.78 and 0.70, for the whole scale, emotional abuse, physical abuse, emotional neglect and emotional neglect, respectively.

#### 2.2.2. Moral Disgust Scale (MD)

The Chinese version of the moral disgust scale was used to measure moral disgust, which was a subscale in the Three-Domain of Disgust Scale originally developed by Tybur et al. (2009). It consists of 8 items, including items such as “A student cheating to get good grades”. Each item was rated on a 7-point Likert-type scale of 0 (not at all) to 6 (strongly disgust). Higher scores means higher moral disgust. In this study, we adapted the subscale into Chinese and the internal consistency of the scale was adequate (α = 0.72).

#### 2.2.3. Wong Law Emotional Intelligence Scale (WLEIS)

The Wong Law Emotional Intelligence Scale [23] is a 16-item questionnaire designed to measure one’s emotional intelligence, divided into four subscales: self-emotion appraisals, others’ emotion appraisals, regulation of emotion, and use of emotion. Sample items included “I always know my friends emotions from their behaviour”. Items were rated on a 5-point Likert-type scale ranging from 1 (strongly disagree) to 5 (strongly agree). In this paper, we used the Chinese revised version, which has been proved to be reliable and valid in China [27,39]. In the current study, the internal consistencies of the four subscales were satisfactory: self-emotion appraisals: 0.77, others’ emotion appraisals: 0.83, regulation of emotion: 0.85, and use of emotion: 0.81. Additionally, the internal consistency of the full scale was 0.89.

#### 2.2.4. Interpersonal Reactivity Index–C (IRI)

The Interpersonal Reactivity Index was originally developed by Davis [40] and consists of four subscales, including perspective-taking (PT), personal distress (PD), fantasy (FS), and empathic concern (EC). There are 22 items in total, such as: “I really get involved with the feelings of the characters in a novel”. It was rated on a 5-point Likert-type scale ranging from 0 (strongly disagree) to 4 (strongly agree). Higher scores account for higher levels of empathy. Additionally, in this study, we used the Chinese version, named Interpersonal Reactivity Index–C, which has been found to have adequate reliability and validity in the Chinese population [41]. In the current study, the internal consistencies of the four subscales were 0.80 for perspective-taking, 0.81 for personal distress, 0.78 for fantasy, and 0.70 for empathic concern. Additionally, the internal consistency of the full scale was also acceptable (α = 0.77).

### 2.3. Data Analysis

We analysed the data using SPSS 22.0 (IBM, Armonk, NY, USA) and AMOS 24.0 (IBM, Chicago, IL, USA). First, descriptive analysis was conducted to show the fundamental information of samples. Second, we used structural equation modelling (SEM) to test the association of variables. The measurement model was first built to test the fit of observed variables to the latent variables. The items in the Moral Disgust Scale were divided into two parcels, and the items in the Childhood Trauma Questionnaire (CTQ), Wong Law Emotional Intelligence Scale (WLEIS) and Interpersonal Reactivity Index–C (IRI) were divided according to their dimensions, serving as observed variables of moral disgust, childhood maltreatment, emotional intelligence and empathy, respectively, in the method of item-to-construct balance [42]. Based on the evidence of the measurement model, we further conducted the structural model analysis. Several indexes were used to assess the model’s goodness: the chi-square fit statistic, comparative fit index (CFI), root-mean-square error of approximation (RMSEA) and standardized root-mean-square residual (SRMR). For the SRMR and RMSEA, values of 0.080 or below indicate acceptable fit. Additionally, for the CFI, values of 0.900 or above indicates the goodness of fit [43]. Besides, the Akaike Information Criterion (AIC) was used for model comparison purposes, and the lower the AIC, the better goodness of fit was deemed [44]. Additionally, expected cross-validation index (ECVI) was also used to evaluate the model’s applicability in different samples. Finally, we conducted bootstrapping procedures to test the mediation effects in this model.

## 3. Results

### 3.1. Measurement Model

The test of proposed measurement model revealed an acceptable fit to the data [χ^2^ _(71, N=968)_ = 483.565, *p* < 0.001; RMSEA = 0.078; SRMR = 0.067; CFI = 0.856]. All observed variables were significantly loaded on corresponding latent variables (*p* < 0.001), which means all latent constructs were adequately measured by their indicators. Additionally, inter-correlations for all variables are presented in Table 1, revealing the significant correlations of latent variables.

### 3.2. Structural Model

First, childhood maltreatment can predict moral disgust directly (Beta = −0.133, *p* < 0.001) (standardized). Then, **Model 1** was constructed to examine our hypothesis. In this model, there is a direct path from childhood maltreatment to moral disgust and two mediators (emotional intelligence and empathy), and in addition, emotional intelligence can also predict empathy. The analysis of **Model 1** indicated an acceptable fit [χ^2^ _(71, N=968)_ = 483.565, *p* < 0.001; RMSEA = 0.078; SRMR = 0.067; CFI = 0.856] (see Table 2). However, two standardized path coefficients were not significant, one is the path from childhood maltreatment to empathy (Beta = −0.013, *p* = 0.532), and the other one is the path from emotional intelligence to moral disgust (Beta = 0.069, *p* = 0.168), these paths were thus constrained to zero. According to these results, we further constructed **Model 2**, which was better than **Model 1** [χ^2^ _(73, N=968)_ = 485.724, *p* < 0.001; RMSEA = 0.076; SRMR = 0.067; CFI = 0.856] (see Table 2). After analysing the correction coefficients, we found acceptable correlations between the error terms for emotional abuse and physical abuse in childhood maltreatment, as well as for the error terms for perspective-taking and personal distress in empathy. Therefore, we constructed **Model 3**. The results revealed that **Model 3** fit best [χ^2^ _(71, N=968)_ =344.258, *p* < 0.001; RMSEA = 0.063; SRMR = 0.058; CFI = 0.905] (see Table 2). As a result, **Model 3** was set as the final model (see Figure 1).

### 3.3. Test of the Mediation Model

We conducted bootstrapping procedures to test the mediation effects in **Model 3**. Two thousand bootstrapping samples were generated from the collected data (N = 968) by random sampling. Table 3 shows the mediating effects and their 95% confidence intervals. Specifically, childhood maltreatment has an indirect effect on empathy through emotional intelligence significantly (95% confidence intervals, (−0.091~−0.031)), emotional intelligence has an indirect effect moral disgust through empathy significantly (95% confidence intervals, (0.069~0.231)), and childhood maltreatment indirectly affects moral disgust via the effect of emotional intelligence on empathy significantly (95% confidence intervals, (−0.138~−0.041)).

## 4. Discussion

This paper discussed the influencing mechanism of moral disgust based on the nature of it and proved the negative link between childhood maltreatment and moral disgust. At the same time, the present study explored the specific mechanism of childhood maltreatment’s influence on moral disgust, from the perspective of emotional development and moral development, respectively, and found that the relationship between them could be explained by the continuous mediating effect of “emotional intelligence-empathy”.

As expected, childhood maltreatment negatively predicts moral disgust, which is consistent with **H1**. This result provided empirical support for the ecological systems theory. As the ecological systems theory indicates, the environment that children grow up in has extensive influence on various aspects [45], including behaviour [46], morality [47], etc. Additionally, moral disgust is an important part of the development of morality. Relative research also has demonstrated that parenting style is crucial to the development and maturity of children’s moral emotion [48], for example, warm parents can not only directly impart moral values to children, but also can give children enough emotional support, thus contributing to children’s moral internalization and emotion socialization, which lay a foundation for the advent of moral disgust [49]. On the contrary, parents who abuse children often adopt ignorant and abusive parenting that is not instrumental to the formation of moral disgust [47]. In conclusion, we proved that childhood maltreatment negatively influenced moral disgust in this study, directly supporting previous studies.

In addition, we further explored the mechanism by which childhood maltreatment affected moral disgust from the perspective of emotional and moral development. Additionally, the findings suggest that childhood maltreatment negatively impacts empathy through emotional intelligence, and thereby impedes the formation of moral disgust. Although the result was inconsistent with **H2** and **H3**, it proved **H4**. We can conclude from results above, first of all, that childhood maltreatment affects empathy through emotional intelligence. It is not surprising that childhood maltreatment hampers the formation of emotional intelligence, which has been demonstrated by previous research [27,50]. The result is also in line with the tripartite model. According to the tripartite model, the family environment could affect children’s emotional ability in the respect of understanding and regulating [16], while childhood maltreatment accompanies a typical adverse home environment, and emotional intelligence is an important ability concerning emotion. It could also be explained that childhood maltreatment leads to more exposure to negative emotions and less social support as children grow up, resulting in difficulties in processing and managing emotions [51] that is reflected in emotional intelligence. Additionally, poor emotional intelligence may further hinder the development of empathy, because the ability to perceive and understand the emotion of other people is also an important part of emotional intelligence [23]. As a result, people with high emotional intelligence tend to be more sensitive to capture others’ feelings and emotions, and more accurate to look inside people’s thoughts. In contrast, poor emotional intelligence may lead to difficulties in generating empathy. Secondly, emotional intelligence influences moral disgust through empathy. It shows that emotional intelligence not only affects empathy, but also further leaves an effect on moral disgust through empathy. The reason may be that empathy is basic for forming moral standards. Previous studies have shown that “Empathy is to moral thought and action what hunger is to the evaluation and consumption of food” [52], so it would be difficult for individuals lacking empathy to develop moral concepts. Therefore, it leads to deficiency in making a correct judgment in moral situations, and thus obstructs the advent of moral disgust. This result also aligns with previous studies, which once again proves the important effects of empathy on various aspects of morality. For example, Decety and Cowell (2014) [31] believe that the emotional, motivational and cognitive components of empathy can all affect moral behaviours, and Mealey (1995) [53] proves that it is the lack of empathy that accounts for the immoral behaviours such as sociopaths’ attacks. Therefore, it is reasonable to infer that empathy is probably not just crucial for moral disgust, but leaves an effect on moral emotion in a broader sense, which is worth exploring further in future research.

Additionally, it is noteworthy that although the results of this study do not support **H2** and **H3**, which are inconsistent with our former expectations, they are reconcilable with the theoretical base and previous studies. On the one hand, the results showed that childhood maltreatment could not directly predict empathy, but revealed that emotional intelligence served as an important role in mediating the relationship between childhood maltreatment and empathy, which was not merely supportive for previous research [54], but also uncovered its mechanism in a deeper level. On the other hand, emotional intelligence cannot directly predict moral disgust, but had an indirect effect on moral disgust through empathy, which is also an effective supplement to previous studies. It once again proves that morality is not the product of pure rationality, but could also be affected by emotion [55], explaining the specific mechanism of emotional intelligence on moral disgust further.

## 5. Limitation

The current study has some limitations that should be considered. First, the study was conducted based on self-report measures that are susceptible to social desirability bias. Second, although we constructed structural equation models that can make preliminary inferences about the causal relationships between variables, longitudinal studies or experimental researches are required to verify the precise causal relationships. Third, the childhood trauma questionnaire used in the study showed low reliability, and future studies could consider developing a more reliable instrument. Finally, the present study recruited participants who were Chinese undergraduates, and the sample representativeness was somewhat limited. Future research should be performed on participants from different age groups and cultural backgrounds.

## 6. Conclusions

In conclusion, based on the nature of moral disgust, the present paper proves that childhood maltreatment has a negative effect on moral disgust, and explores the influencing mechanism between them from the perspective of emotional development and moral development. The results show that childhood maltreatment might affect the development of empathy through emotional intelligence and thus hinder the formation of moral disgust. This study extends previous research on moral disgust and provides a meaningful perspective for the intervention of it. At the same time, it is worth noting that moral disgust is a field of great significance to research, and it is necessary to arouse more relevant studies. As a moral emotion, it not only influences individual behaviour, but also plays a profound role in culture shaping [56]. Emotions are contagious, therefore, when moral disgust permeates the whole social culture, it will bring immeasurable social impact. Prejudice and discrimination against certain groups in different societies are probably related to moral disgust [52]. This paper mainly discusses the factors affecting moral disgust from the individuals and families levels. In the future, more discussions can be carried out from the perspective of society, so as to have a deeper understanding of the causes of moral disgust.

## Figures and Tables

**Figure 1 ijerph-19-10411-f001:**
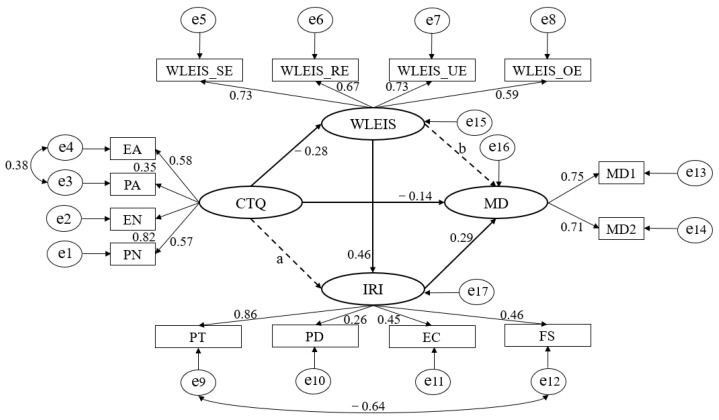
The standardized mediation model.

**Table 1 ijerph-19-10411-t001:** Descriptive statistics and bivariate correlations for all measures.

	M	SD	1	2	3	4
1.CTQ	36.02	9.11	1.000			
2.WLEIS	80.37	12.20	−0.198 ***	1.000		
3.IRI	41.06	4.50	−0.065 **	−0.213 ***	1.000	
4.MD	53.75	10.12	−0.133 ***	0.154 ***	0.185 ***	1.000

Note. ** *p* < 0.01, *** *p* < 0.001.

**Table 2 ijerph-19-10411-t002:** Fit indices of Model 1, Model 2, and Model 3.

	χ^2^	df	CFI	RMSEA	SRMR	AIC	ECVI
Model 1	483.565	71	0.856	0.078	0.067	551.565	0.570
Model 2	485.724	73	0.856	0.076	0.067	549.724	0.568
Model 3	344.258	71	0.905	0.063	0.058	412.258	0.426

Note. CFI, comparative fit index; RMSEA, root-mean-square error of approximation; SRMR, standardized root-mean-square residual; AIC, Akaike information criterion; ECVI, expected cross-validation index.

**Table 3 ijerph-19-10411-t003:** Standardized indirect effects and 95% confidence intervals.

Pathways	Estimate	Lower	Upper
CTQ→WLEIS→IRI	−0.129	−0.091	−0.031
WLEIS→IRI→MD	0.133	0.069	0.231
CTQ→WLEIS→IRI→MD	−0.078	−0.138	−0.041

## Data Availability

The data used for the statistical analysis in the current study are not publicly available due to privacy concerns. Nevertheless, these data are available upon request from the corresponding author.

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
