# Peer review of "Does Childhood Maltreatment Predict Moral Disgust? The Underlying Mediating Mechanisms"

_ijerph, 2022, doi:10.3390/ijerph191610411_

Round 1
Reviewer 1 Report
The study covers an interesting topic, for the developmental psychologists, clinical psychologists, school psychologists.
Moral disgust can be found as an basic concept for the emotional and social development.
I would suggest a better highlight of this concept in the introduction, as well as in the discussion.
The four hypothesis should be presented at the and of the introduction, in order to better follow the red line in-between them.
The discussion brings up more topics/concepts, for exemple the Tripartite Model, which is not presented in the introduction.
Overall, the study set an interesting spot on crucial concepts for the development.
Author Response
Response to Reviewer 1Comments
Reviewer: #1:The study covers an interesting topic, for the developmental psychologists, clinical psychologists, school psychologists.
1.Moral disgust can be found as an basic concept for the emotional and social development.I would suggest a better highlight of this concept in the introduction, as well as in the discussion.
2.The four hypothesis should be presented at the and of the introduction, in order to better follow the red line in-between them.
3.The discussion brings up more topics/concepts, for exemple the Tripartite Model, which is not presented in the introduction.
Overall, the study set an interesting spot on crucial concepts for the development.
Response:
Thank you very much for reviewing this manuscript (ijerph-1851752).
According to your suggestion, we answered the questions and revised the relevant parts of the manuscript. First of all, we displayed the revised contents at the back of each question in italics and noted the numbers of page and line. Second, we marked the revised content in red in our manuscript.
Point 1: Moral disgust can be found as an basic concept for the emotional and social development.I would suggest a better highlight of this concept in the introduction, as well as in the discussion.
Response 1: Thank you for your suggestions.
We have read your comments carefully and realize the importance of highlighting relevant concepts.Therefore, we have added that moral disgust is affected by emotional and social development. The details are as follows:
“Moral disgust, as a moral emotion, is necessarily affected by moral and emotional development.Specifically,emotional development enables individuals to respond to disgusting stimulus with an appropriate emotional reaction and the development of morality reflects the individual's possession of moral concepts and moral attitudes [14].” (Page 2, Line 57 to Line 59.)
Point 2: The four hypothesis should be presented at the and of the introduction, in order to better follow the red line in-between them.
Response 2: Thank you for your suggestions.
We have read your comments carefully and have highlighted the four hypotheses presented at the end of the introduction. The details are as follows:
“(1) childhood maltreatment negatively predicts moral disgust; (2) childhood maltreatment predicts moral disgust negatively through emotional intelligence; (3) childhood maltreatment predicts moral disgust negatively through empathy; (4) childhood maltreatment influences empathy through emotional intelligence, and then negatively predicts moral disgust.” (Page 3, Line 131 to Line 135.)
Point 3: The discussion brings up more topics/concepts, for exemple the Tripartite Model, which is not presented in the introduction.
Response 3: Thank you for your suggestions.
Considering the logical construction of the entire article, we elaborate more on the tripartite model in the introduction, which could also strengthen the theoretical contribution of the current manuscript. The details are as follows:
“Overall, this study probed into the specific mechanism between childhood maltreatment and moral disgust from the perspective of emotional and moral development based on the Tripartite Model, so as to enrich relevant theoretical research about moral disgust and provide suggestions to intervene the development of moral disgust.” (Page 2, Line 64 to Line 68.)
“As implied by the tripartite model of the impact of the family, the family environment in which the child lives plays an important role in the development of the child's emotional competence. As implied by the tripartite model of the impact of the family, the family environment in which the child lives plays an important role in the development of the child's emotional competence.” (Page 2, Line 78 to Line 80.)
“Combining the above theoretical and empirical evidence, the adverse family context in childhood leads to low emotional intelligence and consequently low disgust for immorality.” (Page 2, Line92 to Line 94.)
“As empathy could be considered as an individual's affective responsiveness, it also fits the Tripartite Model's view of childhood maltreatment affecting the development of emotional competence.Thus, empathy may also explain the link between childhood maltreatment and moral disgust, and some indirect evidence helps support our hypothesis.” (Page 3, Line 103 to Line 107.)
Reference:
Morris, A. S.; Silk, J. S.; Steinberg, L.; Myers, S. S.; Robinson, L. R. The Role of the Family Context in the Development of Emotion Regulation. Soc. Dev. 2007, 16 (2), 361–388. https://doi.org/10.1111/j.1467-9507.2007.00389.x.

Reviewer 2 Report
This paper examined he concept of moral disgust as a moral emotion using a college student sample from China. The problem of this article is on childhood maltreatment to determine emotional and moral development. The literature is clear and well articulated.
However, it is suggested that a theory be selected to help the reader understand the connections you are trying to establish from this study.
It is unclear about the demographics of the participants other than their gender and that they are from China. Please reveal any other demographics, such as age.
The design of the study needs to be articulated in the methods section.
The instruments were effective with good alpha reliabilities.
The data analysis section provides sufficient information on the types of analyses that were conducted.
Results were conducted effectively. The models were compared based on effective, thereby achieving the best possible fit.
The findings were discussed effectively with sufficient context.
Please include the low reliability found in the study (.65) as a limitation. Suggest another alternative Childhood Trauma instrument with better reliability results, or suggest the need to develop more reliable measures to assess this construct.
There are several writing and inconsistency errors on this paper, please have someone proof the paper. Examples, reduce the capital letter, “First, The” in line 313,” for example.
Author Response
Response to Reviewer 2 Comments
Reviewer: #2: This paper examined he concept of moral disgust as a moral emotion using a college student sample from China. The problem of this article is on childhood maltreatment to determine emotional and moral development. The literature is clear and well articulated.
1.However, it is suggested that a theory be selected to help the reader understand the connections you are trying to establish from this study.
2.It is unclear about the demographics of the participants other than their gender and that they are from China. Please reveal any other demographics, such as age.
3.The design of the study needs to be articulated in the methods section.
The instruments were effective with good alpha reliabilities.
The data analysis section provides sufficient information on the types of analyses that were conducted.
Results were conducted effectively. The models were compared based on effective, thereby achieving the best possible fit.
The findings were discussed effectively with sufficient context.
4.Please include the low reliability found in the study (.65) as a limitation. Suggest another alternative Childhood Trauma instrument with better reliability results, or suggest the need to develop more reliable measures to assess this construct.
5.There are several writing and inconsistency errors on this paper, please have someone proof the paper. Examples, reduce the capital letter, “First, The” in line 313,” for example.
Response:
Thank you very much for reviewing this manuscript (ijerph-1851752).
According to your suggestion, we answered the questions and revised the relevant parts of the manuscript. First of all, we displayed the revised contents at the back of each question in italics and noted the numbers of page and line. Second, we marked the revised content in red in our manuscript.
Point 1: However, it is suggested that a theory be selected to help the reader understand the connections you are trying to establish from this study.
Response 1: Thank you for your suggestions.
It is true that the theoretical framework is crucial for the reader to better understand this study. Therefore, according to your suggestion, we have elaborated more about the model's theoretical background in the introduction section to better support the supposed model.
The current study explores the relationship between childhood maltreatment and moral disgust, as well as the potential mediating role of emotional intelligence and empathy in this relationship. We tried to construct associations between variables according to the Tripartite Model of the impact of family. The Tripartite Model argues that the family context plays an important role in the development of a child's emotional competence. On the one hand, childhood maltreatment is a typical adverse family context. On the other hand, emotions could affect the development of moral competence. Therefore, childhood maltreatment(family context) may influence moral disgust by affecting emotional intelligence as well as empathy(emotional competence). Then, we supported the theoretical assumptions with some empirical evidence.The details are as follows:
“Overall, this study probed into the specific mechanism between childhood maltreatment and moral disgust from the perspective of emotional and moral development based on the Tripartite Model, so as to enrich relevant theoretical research about moral disgust and provide suggestions to intervene the development of moral disgust.” (Page 2, Line 64 to Line 68.)
“As implied by the tripartite model of the impact of the family, the family environment in which the child lives plays an important role in the development of the child's emotional competence. As implied by the tripartite model of the impact of the family, the family environment in which the child lives plays an important role in the development of the child's emotional competence.” (Page 2, Line 78 to Line 80.)
“Combining the above theoretical and empirical evidence, the adverse family context in childhood leads to low emotional intelligence and consequently low disgust for immorality.” (Page 2, Line92 to Line 94.)
“As empathy could be considered as an individual's affective responsiveness, it also fits the Tripartite Model's view of childhood maltreatment affecting the development of emotional competence.Thus, empathy may also explain the link between childhood maltreatment and moral disgust, and some indirect evidence helps support our hypothesis.” (Page 3, Line 103 to Line 107.)
Point 2: It is unclear about the demographics of the participants other than their gender and that they are from China. Please reveal any other demographics, such as age.
Response 2: Thank you for your suggestions.
After having read your comments carefully, we have revised related contents. The details are as follows:
“(619 females and 349 males; M = 19.07, SD = 1.55, from 17 to 26 years old).” (Page 3, Line 141 to Line 143.)
Point 3: The design of the study needs to be articulated in the methods section.
Response 3: Thank you for your suggestions.
In this study,firstly,both cluster sampling and random sampling were used to recruit participants.. Secondly,participants are invited to complete the questionnaire required for the study, which takes approximately 40 minutes and the informed consent form was provided to all participants before administration of the survey. Finally, the data included in the analysis excluded questionnaires that were not completed or completed in a mischievous way. We have revised the relevant content to give the reader a more detailed understanding of our study design. The details are as follows:
“After excluding data from participants who did not complete the questionnaire or completed it in a mischievous way, data from a total of 968 participants were included in the final analysis.The youngest participant was 17 years old and the oldest participant was 26 years old.The mean age of them was 19.07 years old (SD = 1.55), and 64% (n = 619) were women. Participants are invited to complete the questionnaire required for the study, which takes approximately 40 minutes.” (Page 3, Line 139 to Line 144.)
Point 4: Please include the low reliability found in the study (.65) as a limitation. Suggest another alternative Childhood Trauma instrument with better reliability results, or suggest the need to develop more reliable measures to assess this construct.
Response 4: Thank you for your suggestions.
As you mentioned, the low reliability of the measurement instrument is a limitation of the current study, and we have written about it in the limitations section. The details are as follows:
“Third, the childhood trauma questionnaire used in the study showed low reliability, and future studies could consider developing a more reliable instrument.” (Page 8, Line 328 to Line 330.)
Point 5: There are several writing and inconsistency errors on this paper, please have someone proof the paper. Examples, reduce the capital letter, “First, The” in line 313,” for example.
Response 5: Thank you for your suggestions.
Sorry for the error caused by our carelessness, we have read the entire article critically and revised the error.
Reference:
Morris, A. S.; Silk, J. S.; Steinberg, L.; Myers, S. S.; Robinson, L. R. The Role of the Family Context in the Development of Emotion Regulation. Soc. Dev. 2007, 16 (2), 361–388. https://doi.org/10.1111/j.1467-9507.2007.00389.x.
